# Analysis of unusual and signature APOBEC-mutations in HIV-1 *pol* next-generation sequences

**Philip L. Tzou** [1]*, **Sergei L. Kosakovsky Pond**[2], **Santiago Avila-Rios**[3], **Susan P. Holmes**[4], **Rami Kantor**[5], **Robert W. Shafer**[1]*

**1** Division of Infectious Diseases, Department of Medicine, Stanford University, Stanford, CA, United States of America, **2** Institute for Genomics and Evolutionary Medicine, Temple University, Philadelphia, PA, United States of America, **3** Centre for Research in Infectious Diseases, National Institute of Respiratory Diseases, Tlalpan, Mexico City, Mexico, **4** Department of Statistics, Stanford University, Stanford, CA, United States of America, **5** Division of Infectious Diseases, Department of Medicine, Brown University, Providence, RI, United States of America

* philiptz@stanford.edu (PLT); rshafer@stanford.edu (RWS)

## Abstract

### Introduction

At low mutation-detection thresholds, next generation sequencing (NGS) for HIV-1 genotypic resistance testing is susceptible to artifactual detection of mutations arising from PCR error and APOBEC-mediated G-to-A hypermutation.

### Methods

We analyzed published HIV-1 *pol* Illumina NGS data to characterize the distribution of mutations at eight NGS mutation detection thresholds: 20%, 10%, 5%, 2%, 1%, 0.5%, 0.2%, and 0.1%. At each threshold, we determined proportions of amino acid mutations that were unusual (defined as having a prevalence <0.01% in HIV-1 group M sequences) or signature APOBEC mutations.

### Results

Eight studies, containing 855 samples, in the NCBI Sequence Read Archive were analyzed. As detection thresholds were lowered, there was a progressive increase in the proportion of positions with usual and unusual mutations and in the proportion of all mutations that were unusual. The median proportion of positions with an unusual mutation increased gradually from 0% at the 20% threshold to 0.3% at the 1% threshold and then exponentially to 1.3% (0.5% threshold), 6.9% (0.2% threshold), and 23.2% (0.1% threshold). In two of three studies with available plasma HIV-1 RNA levels, the proportion of positions with unusual mutations was negatively associated with virus levels. Although the complete set of signature APOBEC mutations was much smaller than that of unusual mutations, the former outnumbered the latter in one-sixth of samples at the 0.5%, 1%, and 2% thresholds.

**Data Availability Statement:** All sequences referenced by this manuscript are previously published on NCBI Sequence Read Archive (BioProject Accessions: PRJNA340290,

PRJNA486832, PRJNA517147, PRJNA384904, PRJNA448668, PRJDB3502, and PRJNA531904) and Zenodo (doi:10.5281/zenodo.44921).

**Funding:** RK is supported by a grant from the National Institutes of Health (NIH/NIAID P30AI042853). PLT and RWS are also supported by a grant from the National Institutes of Health (NIH/NIAID R24AI136618) (https://www.niaid.nih.gov/). The funder doesn't play any role in the study design, data collection and analysis, decision to publish, or preparation of the manuscript.

**Competing interests:** The authors have declared that no competing interests exist.

## Conclusions

The marked increase in the proportion of positions with unusual mutations at thresholds below 1% and in samples with lower virus loads suggests that, at low thresholds, many unusual mutations are artifactual, reflecting PCR error or G-to-A hypermutation. Profiling the numbers of unusual and signature APOBEC *pol* mutations at different NGS mutation detection thresholds may be useful to avoid selecting a threshold that is too low and poses an unacceptable risk of identifying artifactual mutations.

## Introduction

Next-generation sequencing (NGS) is increasingly performed for HIV-1 genotypic resistance testing [1]. However, low levels of plasma viremia and/or inefficient RNA extraction, or reverse transcription may result in a low number of amplifiable cDNA templates. In such scenarios, much of the observed variability in an NGS sequence may reflect PCR error rather than authentic viral mutations [2–4]. Since PCR errors are not subject to selective forces exerted during virus evolution, we have hypothesized that the presence of large numbers of unusual and likely deleterious mutations at an NGS mutation detection threshold suggests the threshold is too low [5–8].

NGS is also more likely than Sanger sequencing to detect low frequency APOBEC-mediated G-to-A hypermutation [9–11]. APOBEC-mediated G-to-A hypermutation can be detected if plasma samples are contaminated with proviral DNA templates, which are enriched for defective viruses [12], or if defective hypermutated virus genomes are successfully packaged and released from cells. Hypermutated viruses are unlikely to be functional because they often contain premature stop codons and mutations at highly conserved residues [9–11, 13]. Therefore, the detection of drug-resistance mutations (DRMs) that could be caused by APOBEC in viruses with evidence for G-to-A hypermutation has questionable clinical significance.

In this study, we systematically analyze HIV-1 *pol* NGS data from eight published studies to characterize the distribution of unusual mutations and mutations suggestive of APOBEC-mediated G-to-A hypermutation at different NGS mutation detection thresholds.

## Methods

### NGS datasets, FASTQ files, and codon frequency tables

We searched the NCBI Sequence Read Archive BioProject Library and other public repositories to identify NGS data sets of HIV-1 *pol* meeting the following criteria: (i) sequencing was performed on Illumina instruments; (ii) samples contained at least 10 clinical specimens; (iii) samples were from plasma HIV-1 RNA rather than proviral DNA; and (iv) samples required PCR amplification (i.e., were not from metagenomic studies). The publications associated with these datasets were reviewed to retrieve the following information for each sample: plasma HIV-1 RNA level, volume of plasma submitted for RNA extraction, number of sequencing reads, and methods of RNA extraction, RT-PCR, and library preparation. Datasets for which there was no associated publication were excluded (Fig 1).

We extended the HYDRA pipeline [14] to generate a codon frequency table from each FASTQ file. Briefly, we filtered reads with fewer than 100 nucleotides or a mean quality (phred or q) score <30 (predicted error rate 1 in 1000). We then aligned the filtered reads to the HXB2 *pol* nucleotide sequence using BOWTIE 2 [15] with the default HYDRA parameters.

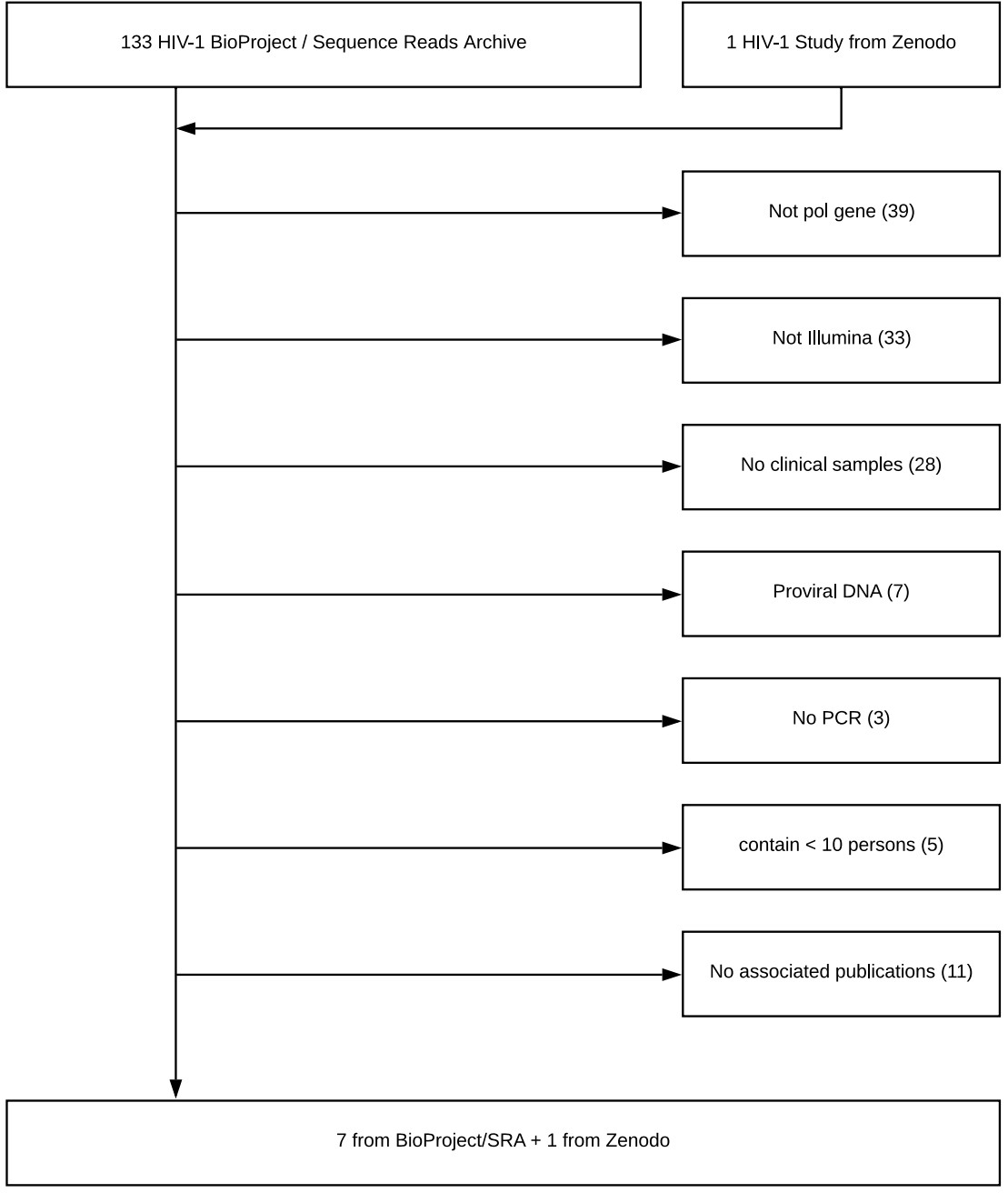

**Fig 1. Flowchart of study selection process.** Of 133 studies in the NCBI BioProject Sequence Read Archive as of August 1, 2019, seven met study selection criteria. One additional study was identified in the research upload site Zenodo (https://zenodo.org/).

Individual nucleotides with a q score lower than 30 were not counted. The consensus sequence for each sample was submitted to the HIVDB drug resistance interpretation program to impute its subtype. Positions covered with $\geq$1000 remaining reads were used to populate a codon frequency table with five fields:

1. Gene–protease (PR), reverse transcriptase (RT), or integrase (IN).

2. Amino acid position.

3. Number of reads for the position (coverage).

4. Sequenced codon.

5. Number of reads containing the codon.

   Each codon frequency table was annotated with the following seven additional fields:

1. The codon's translation. This is usually an amino acid but can also be a stop codon, an insertion (indicated by more than one amino acid), a deletion, or a frame shift if the codon does not contain a multiple of three bases).

2. The fraction of reads containing the codon.

3. The prevalence of the corresponding amino acid in group M sequences in HIVDB.

4. The prevalence of the codon in group M sequences in HIVDB.

5. Whether the amino acid encodes a drug-resistance mutation (DRM).

6. Whether the amino acid is a signature APOBEC mutation.

7. Whether the amino acid is a DRM that could also be caused by APOBEC.

## Unusual mutations

The procedure by which HIVDB mutation prevalence data was determined is documented in a GitHub repository (https://github.com/hivdb/hivfacts). The group M mutation prevalence data was derived from direct PCR ("population based") plasma virus Sanger sequences in HIVDB, from which poor quality sequences had been filtered. The prevalence of an amino acid at a position was calculated by dividing the number of occurrences of all non-mixed amino acids by the total number of sequences containing the position. Each mutation was counted once per individual. Unusual mutations were defined as mutations with a group M prevalence <0.01% and not on the list of known DRMs (i.e., without an HIVDB mutation penalty score).

Overall, 4,651 (23.4%) of the 19,887 possible amino acids in the 947 protease, RT, and integrase positions were classified as usual and 15,236 (76.6%) were classified as unusual. The usual amino acids included the 947 (20.4%) amino acids constituting the Los Alamos National Laboratories group M consensus [16] and 3,704 non-consensus amino acids. The 3,704 non-consensus usual amino acids included 2,750 (74.2%) with one nucleotide difference from the group M consensus, 867 (23.4%) with two nucleotide differences, and 87 (2.3%) with three nucleotide differences. Among the unusual mutations, 3,133 (20.6%) had one nucleotide difference from the group M consensus, 8,503 (55.8%) had two nucleotide differences, and 3,600 (23.6%) had three nucleotide differences.

## Signature APOBEC mutations

Amino acid mutations consistent with APOBEC-mediated G-to-A hypermutation were identified using a modification of a previously described procedure [6]. Briefly, we found all positions in PR, RT, and IN for which >50% of amino acids at a position represented a potential target for APOBEC3F (5'GA dinucleotides) or APOBEC3G (5'GG dinucleotides). We then identified all amino acid mutations that would result from APOBEC3F or 3G editing of these potential targets. Each of the resulting APOBEC-context mutations was then examined for its

prevalence in group M sequences and for its association with stop codons or active site mutations. Stop codons result from APOBEC3G editing of tryptophan (W): `TGG` → `TAG` or `TGG` → `TGA`, when tryptophan is followed by an amino acid beginning with an `A` or `G`. Active site mutations in PR (D25N), RT (D110N, D185N, and D186N), and IN (D64N, D116N, and E152K) result from APOBEC3F editing of aspartic acid `GAC/T (D)` → `AAC/T (N)` or glutamic acid `GAA/G (E)` → `AAA/G (K)`.

APOBEC-context mutations that met the following criteria were considered signature APOBEC mutations: (i) they occurred at a prevalence <0.1% or at a prevalence <0.5% if they occurred frequently in sequences with stop codons or active site mutations and (ii) they were not known DRMs. Overall, we identified 296 signature APOBEC mutations including 45 in PR, 154 in RT, and 97 in IN. Based on a previous study [6] and a comparison with the LANL Hypermut program [17], we determined that *pol* genes containing three or more signature APOBEC mutations in were likely to have undergone APOBEC-mediated G-to-A hypermutation (S1 Text).

Overall, 175 (59.1%) of the 296 signature APOBEC mutations were also unusual (i.e., prevalence < 0.01%). The remaining signature APOBEC mutations, which had a prevalence ranging from 0.01% to 0.4%, were classified based on their genetic context, their rarity, and their strong association with stop codons and active site mutations. In contrast, just 1.2% of the 15,236 unusual mutations were also signature APOBEC mutations.

## Statistical analysis

We calculated the proportion of amino acid positions with usual mutations, unusual mutations, and signature APOBEC mutations at eight NGS mutation detection thresholds. Usual mutations were defined as differences from the subtype B consensus sequence that were not unusual. We also calculated the proportion of all mutations that were unusual (number of unusual mutations / total number of mutations) at these same thresholds. The eight mutation thresholds began at 20%, which is often considered the limit of detection of mixed bases for Sanger sequencing, with each subsequent value approximately two-fold lower than the previous threshold: 10%, 5%, 2%, 1%, 0.5%, 0.2%, and 0.1%. Such "round" thresholds are commonly used in manuscripts performing NGS data interpretation, and are meant to serve as representative values spanning the realistic range used by researchers.

Pearson correlation coefficient (r) was used to quantify the association between a sample's (i) virus load and proportion of positions with usual or unusual mutations; and (ii) median number of sequence reads per position and proportion of positions with usual or unusual mutations.

## Results

### NGS datasets

Eight studies containing 855 samples from 821 persons met the inclusion criteria [18–25] (Fig 1 and Table 1). These samples included 693 PR, 700 RT, and 449 IN NGS sequence sets. Of the RT samples, 209 encompassed all 560 amino acid positions. Ninety percent of the remaining samples encompassed at least the first 240 amino acid positions. Subtype B accounted for 606 (70.9%) of samples. Subtypes A, C, CRF01_AE, and CRF02_AG were the most common non-B subtypes, accounting for 224 (26.2%) of sequences. Plasma HIV-1 RNA levels were available for all sequenced samples in three studies [21, 23, 24].

Table 2 summarizes experimental parameters for each study. Most studies used 0.4 to 1.0 ml of plasma, high-fidelity RT and PCR enzymes, and nested PCR. However, the specific extraction protocols and enzymes used for PCR amplification varied. Amplicon sizes also

**Table 1. Published studies with available HIV-1 pol NGS datasets.**

| Author (Yr) | Title | # Samples[1] | Genes | VL[2] | Region | ARV Status | Subtypes[3] |
|---|---|---|---|---|---|---|---|
| Avila-Rios (2016)[23] | HIV Drug Resistance in Antiretroviral Treatment-Naïve Individuals in the Largest Public Hospital in Nicaragua, 2011–2015 | 255 | PR/RT | Yes | Nicaragua | Naive | B (99.6%) |
| Moscona (2017)[19] | Comparison between next-generation and Sanger-based sequencing for the detection of transmitted drug-resistance mutations among recently infected HIV-1 patients in Israel, 2000–2014 | 78 | PR (76); RT (77); IN (30) | No | Israel | Naïve | B (59%); C (22%); A (14%) |
| Huber (2016) [24] | MinVar: A rapid and versatile tool for HIV-1 drug resistance genotyping by deep sequencing | 33 | PR/RT (33); IN (13) | Yes | Switzerland | NA | B (67%), A (9%), C (9%), CRF02_AG (9%) |
| Nguyen (2018)[20] | Prevalence and clinical impact of minority resistant variants in patients failing an integrase inhibitor-based regimen by ultra-deep sequencing | 134 | IN | No | France | Treated | B (60%), CRF02_AG (26%) |
| Dalmat (2018)[25] | Limited marginal utility of deep sequencing for HIV drug resistance testing in the age of integrase inhibitors | 112 | PR (93); RT (94); IN (38) | No | U.S. | Treated | B (95%), C (4%), D (1%) |
| Jair (2019) [18] | Validation of publicly-available software used in analyzing NGS data for HIV-1 drug resistance mutations and transmission networks in a Washington, DC, cohort. | 42 | PR (34), RT (41), IN (33) | No | U.S. | Treated | B (98%), CRF02_AG (2%) |
| Ode (2015) [21] | Quasispecies Analyses of the HIV-1 Near-full-length Genome With Illumina MiSeq | 92 | PR/RT/IN | Yes | Japan | Treated | B (61%), CRF01_AE (11%), C (11%), CRF02 (9%) |
| Telele (2019) [22] | Pretreatment drug resistance in a large countrywide Ethiopian HIV-1C cohort: a comparison of Sanger and high-throughput sequencing. | 109 | PR/RT/IN | No | Ethiopia | Naive | C (99%), D (1%) |

[1] Ode 2015 contained 92 samples from 58 persons.

[2] Virus load (VL) data was available for all samples in three studies. In Telele 2019, virus load data was available for a small subset of patients.

[3] Samples with uncommon subtypes are not shown.

varied from 750 to 4,400 bp. Across all studies, the median coverage per position was 18,275, with a 5% to 95% range of 2,944 to 81,184.

## Usual and unusual mutations at different NGS mutation detection thresholds

**Pooled data from all datasets.** Fig 2 depicts the proportion of positions with usual (panel A) and unusual (panel B) mutations and the proportion of all mutations that were unusual (number of unusual mutations / total number of mutations, panel C) as a function of mutation detection threshold in pooled samples for all studies.

The median proportion of positions with a usual mutation increased from 5.2% to 11.6% between the 20% and 0.5% thresholds then began doubling to 23.6% at the 0.2% threshold and to 47.2% at the 0.1% thresholds.

The median proportion of positions with an unusual mutation increased from 0% to 0.3% between the 20% and 1% thresholds but then began increasing about four-fold to 1.3% at the 0.5% threshold, 6.9% at the 0.2% threshold, and 23.2% at the 0.1% thresholds.

The median proportion of mutations that were unusual increased from 0% to 1.1% between the 20% and 2% threshold but then jumped to 4.2% at the 1% threshold, 12.0% at the 0.5% threshold, and to 25.1%, and 33.9% respectively at the 0.2% and 0.1% thresholds.

There was a weak but statistically significant relationship between the $\log_{10}$ of the number of sequence reads (i.e., coverage) and the number of usual mutations (correlation coefficient $r$ between 0.21 and 0.24, p<0.001) at the 1%, 2%, 5%, 10%, and 20% thresholds (S1 Fig). The

**Table 2. Sample preparation methods and median number of sequence reads in eight studies of HIV-1 *pol* NGS.**

| Author (Yr) | RNA Extraction[1] | RT / PCR Enzymes[1] | PCR Product Size (bp)[2] | Sequence Length (5% to 95% Range) | Median # Reads (5% to 95% Range) |
|---|---|---|---|---|---|
| Avila-Rios (2016) | QIAamp Viral RNA Mini Kit (1 ml plasma) | SuperScript III OneStep RT PCR followed by Platinum Taq DNA polymerase | 1,592 | 436 (432–440) | 30,079 (10,721–53,845) |
| Moscona (2017) | NucliSENS easyMAG (500 ul plasma) | NA | 1,800 | 436 (313–603) | 5,652 (1,306–11,359) |
| Huber (2016) | NucliSENS easyMAG (500 ul plasma) | PrimeScript One-Step RT PCR Kit followed by Phusion HotStart II HF polymerase | 3,500 | 440 (423–947) | 72,712 (11,486–107,307) |
| Nguyen (2018) | NucliSENS easyMAG (1 ml plasma) | Transcriptor One-Step RT-PCR followed by QS High Fidelity PCR Kit | 763 | 232 (227–235) | 13,878 (8,788–54,314) |
| Dalmat (2018) | Boom silica (400 ul plasma) | GeneAmp RNA PCR Kit | 1,306 (PR/RT) 1,306 (IN) | 358 (318–713) | 5,718 (1,781–17,979) |
| Jair (2019) | QIAamp Viral RNA Mini Kit (150 ul plasma) | HiFi Taq DNA polymerase | 883 (PR), 652 (RT), 1000+ (IN) | 669 (228–771) | 41,833 (2,681–131,625) |
| Ode (2015) | Magnapure compact NA isolation kit (200–400 ul plasma) | PrimeScript I high Fidelity One Step RT-PCR Kit followed by PrimeSTAR GXL DNA Polymerase | 2,700 | 947 (947–947) | 13,322 (4,453–30,279) |
| Telele (2019) | QIAamp Viral RNA Mini Kit (150 ul plasma) | HiFi Taq DNA polymerase | 4,360 | 947 (947–947) | 31,288 (16,467–96,237) |

Footnote:

[1]Reagent manufacturers: QIAamp Viral RNA Mini Kit (QIAGEN); NucliSENS easyMAG (bioMerieux Clinical Diagnostics); Magnapure compact NA isolation kit (Roche Life Sciences); SuperScript III OneStep RT PCR (Invitrogen); PrimeScript One-Step RT PCR Kit (Takara, Kusatsu, Japan); Phusion HotStart II HF polymerse (ThermoFisher); Transcriptor One-Step RT-PCR (Roche); QS High Fidelity PCR Kit (New England Biolabs); GeneAmp RNA PCR Kit (Perkin-Elmer); HiFi Taq DNA polymerase (Takara; Mountain View, CA, US). For Ode 2015, products from three separate PCR reactions were pooled.

[2]PCR product sizes were estimated from the HXB2 coordinates provided for the first round of PCR. For Jair 2019, it was not possible to precisely determine the size of the integrase (IN) first-round PCR product.

relationship between the number of sequence reads and the number of unusual mutations was much weaker with *r* between 0.13 and 0.17 at the 1%, 2%, and 5% thresholds (S2 Fig).

**Inter- and intra-study variation.** Fig 3A and 3B plot the median proportion of positions with usual and unusual mutations, respectively, at the eight NGS mutation detection thresholds for pooled samples within each of the eight studies. Fig 3C plots the median proportion of mutations that were unusual at each threshold for each study. These figures indicate that there were clear inter-study differences in the distributions of usual and unusual mutations, particularly at thresholds below 1%. The eight studies visually clustered into three groups at the three lowest thresholds. Two studies had the lowest numbers of both usual and unusual mutations, and of the proportions of mutations that were unusual [21, 22]. Four studies had intermediate values for these three metrics [18, 20, 24, 25], and two had higher values [19, 23].

One of the two studies with the lowest numbers of unusual mutations pooled PCR product from three separate reactions, potentially lowering the impact of artifacts propagated per reaction [21]. However, because of the many inter-study differences in sample characteristics and laboratory procedures, we were otherwise unable to assess whether specific sample

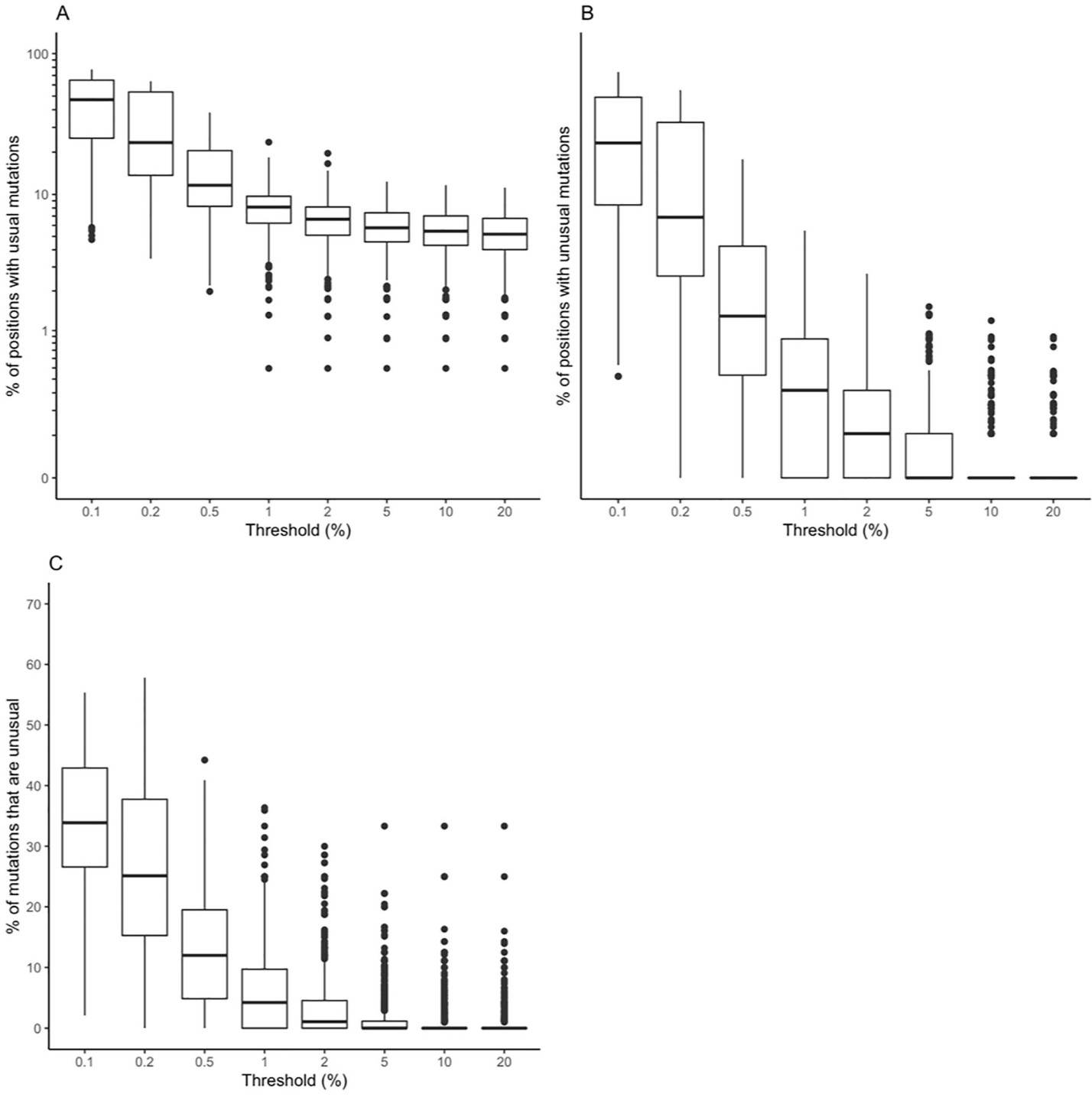

**Fig 2.** Boxplots demonstrating the distribution in the the proportion of positions with usual mutations (A), the proportion of positions with unusual mutations (B), and the proportion of mutations that were unusual (number of unusual mutations / [number of usual mutations + number of unusual mutations]) (C) at eight NGS mutation detection thresholds for pooled samples (n = 855) from eight published studies.

characteristics or laboratory procedures were responsible for observed differences in the proportion of positions with usual and unusual mutations.

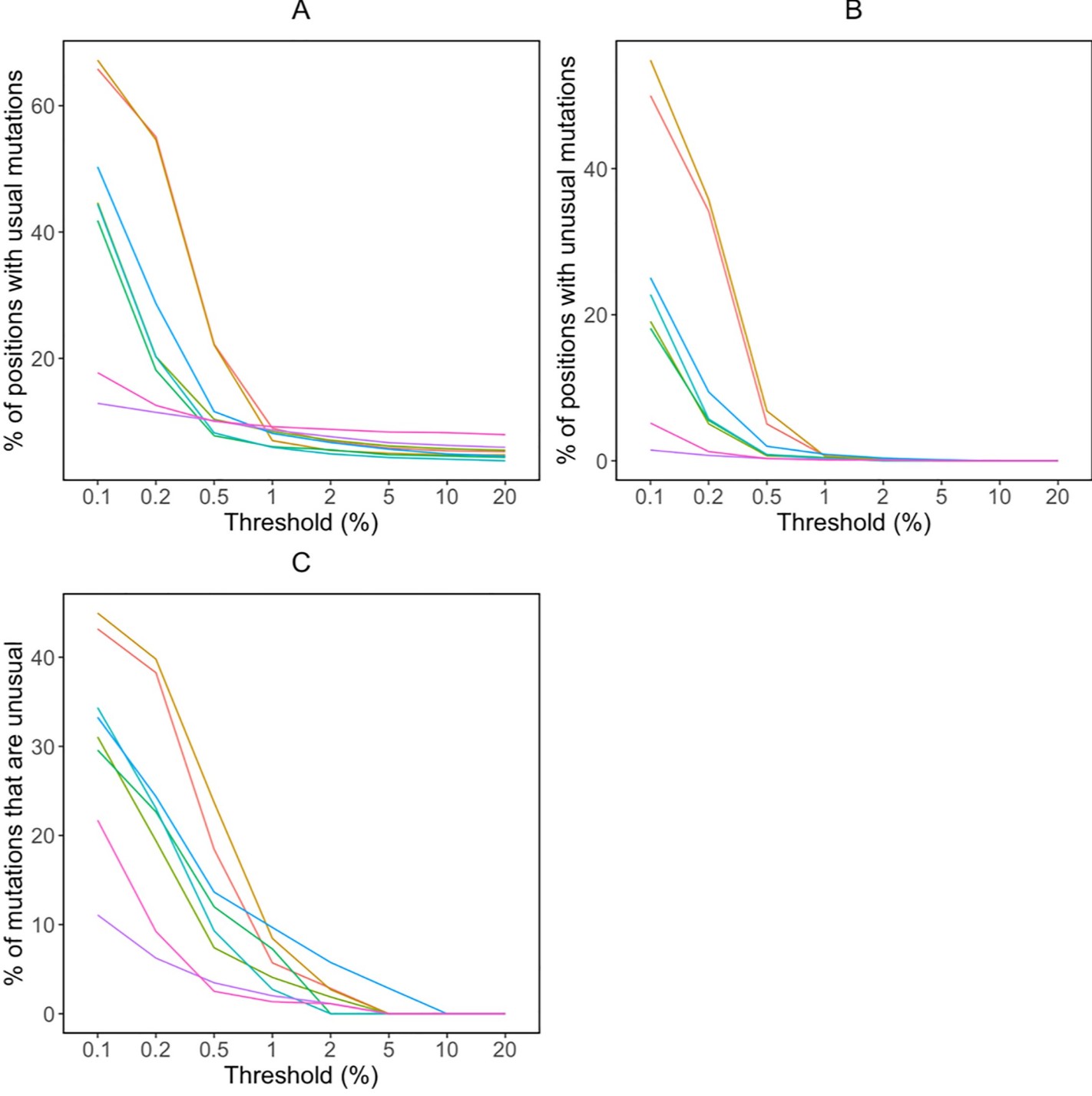

**Fig 3.** Median proportions of positions with usual mutations (A), proportions of positions with unusual mutations (B), and proportions of mutations that were unusual (number of unusual mutations / [number of usual mutations + number of unusual mutations]) (C) at eight NGS mutation detection thresholds for the pooled 855 samples in eight published datasets: light red (#FF6C67) [23], gold (#D79400) [19], lime green (#6CB100) [24], jade (#00C25C) [20], egg blue (#00C3C6) [25], sky blue (#00ABFF) [18], purple (#D475FF) [21], rose (#FF4ED1) [22].

There was marked heterogeneity in the distribution of unusual mutations at different thresholds within each study (S3 Fig). For example, at the 1% threshold, the highest number of

unusual mutations in each study was generally about five times higher than the median number of unusual mutations.

**Association of unusual mutations with virus load.** Plasma HIV-1 RNA levels were available for samples from three of the eight studies [21, 23, 24]. The percent of positions with unusual mutations was significantly higher in samples with lower virus loads in two of the three studies [23, 24], particularly at the 0.5%, 1%, 2%, and 5% thresholds (Fig 4A and 4B). This relationship was not detectable either at the very low thresholds of 0.1% and 0.2%, which contained high numbers of both usual and unusual mutations, or at the 10% and 20% thresholds, which contained few unusual mutations. There was generally no significant relationship between the proportion of positions with usual mutations and the sample's virus load (S4 Fig).

**Number of APOBEC mutations at different NGS mutation detection thresholds.** Fig 5 shows the proportion of positions with signature APOBEC mutations at each threshold. At the 20%, 10%, 5%, 2%, and 1% thresholds, the median proportion of positions with a signature APOBEC mutation was 0. However, at the 2% and 1% thresholds, 46 and 113 samples, respectively, contained three or more signature APOBEC mutations; given the average sequence length, this corresponds to approximately 0.9% of positions.

Although the complete set of signature APOBEC mutations (n = 296) was much smaller than the complete set of unusual mutations that were not signature APOBEC mutations (n = 14,940), signature APOBEC mutations outnumbered non-APOBEC unusual mutations in 16.0%, 17.3%, and 13.2% of samples at the 0.5%, 1%, and 2% thresholds, but in just 2.8% to 7.5% of samples at the remaining thresholds (Fig 6).

## Discussion

Through a meta-analysis of NGS *pol* data from 855 samples in eight published studies, we found that as the mutation detection threshold was lowered, there was a progressive increase in the proportion of sequence positions with both usual and unusual mutations and in the proportion of mutations that are unusual. The proportion of positions with unusual mutations increased gradually from 0% to 0.3% between the 20% and 1% thresholds and then exponentially to 1.3%, 6.9%, and 23.2% at the 0.5%, 0.2%, and 0.1% thresholds, respectively. Similarly, the proportion of mutations that were unusual increased gradually from 0% to 1.1% between the 20% and 2% threshold but then exponentially to 4.2%, 12.0%, and 25.1%, at the 1%, 0.5%, and 0.2% thresholds, respectively.

The marked increase in the proportion of positions with unusual mutations and in the proportion of mutations that were unusual at detection thresholds below 1% suggests that many of the mutations at low thresholds resulted from processes other than virus replication such as PCR error and APOBEC-mediated hypermutation. Although HIV-1 RT produces approximately one random nucleic acid mutation per genome each replication cycle [26, 27], the fitness costs of most random nonsynonymous mutations are high. Therefore, viruses containing many, if not most, random mutations are rarely compatible with levels of replication that would yield viruses detectable at levels significantly higher than the background error rate [28]. In contrast, PCR errors and APOBEC-mediated G-to-A hypermutation can result in mutations detectable at levels that would not be consistent with virus replication.

In two of the studies for which plasma HIV-1 RNA levels were available, the proportion of positions with unusual mutations was inversely related to virus levels at the 0.5%, 1%, 2%, and 5% thresholds. A plausible explanation for this pattern is that samples with lower virus loads yield fewer cDNA molecules and that, in these samples, a greater amount of sequence variation results from PCR amplification rather than from HIV-1 replication.

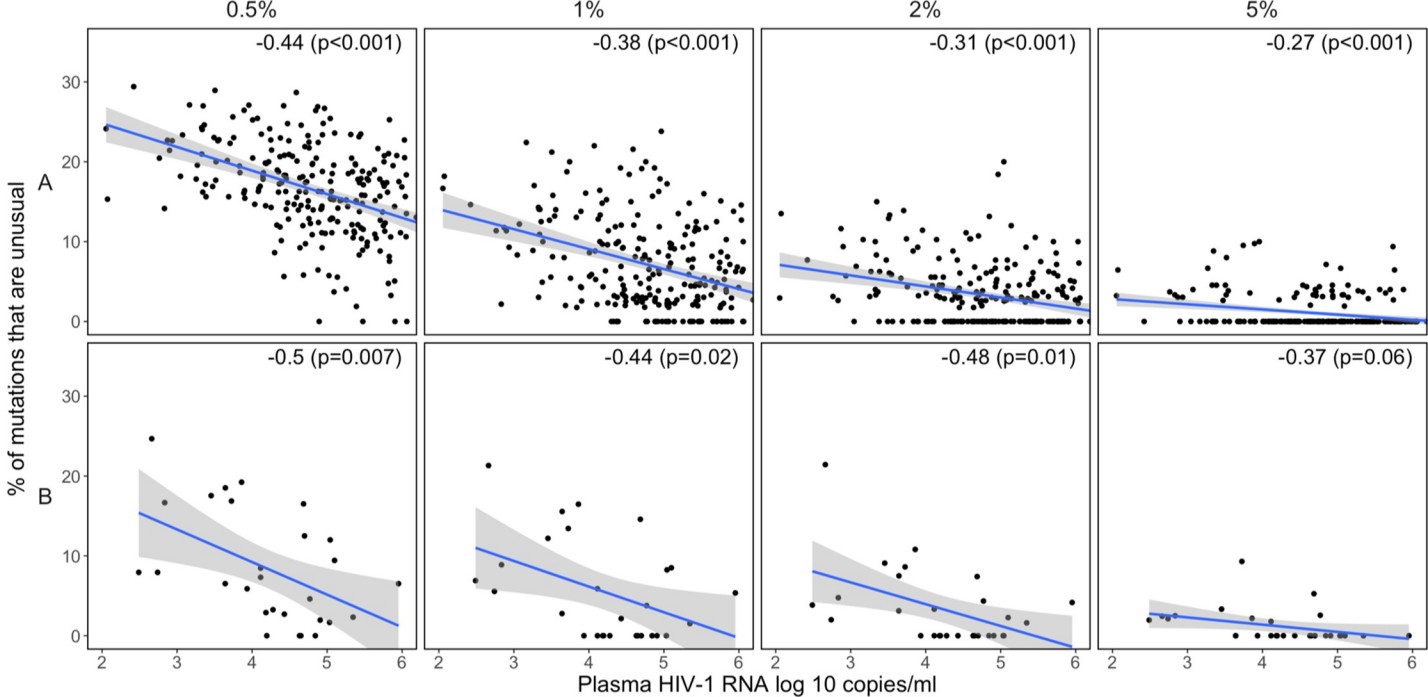

**Fig 4. Scatter plots demonstrating the relationship between virus load (plasma HIV-1 RNA log copies/ml) and the proportion of positions with unusual mutations at four NGS mutation detection thresholds in two of the three studies for which virus load data were available [21, 23, 24]: study A [23], study B [24].** The upper-right hand corner of each plot contains the Pearson correlation coefficient (r) and its associated p value.

Illumina sequence errors were also likely to have contributed to sequence artifact but only in those samples for which the read coverage was too low to achieve the redundancy required to prevent random machine errors from being detected at low thresholds. Indeed, over the complete dataset, the median coverage per position was 18,275 and 95% of positions had a coverage of nearly 3,000 reads. Thus for 95% of samples, machine error would have required the same random error to occur at least three times to result in detectable sequence artifacts at the 0.1% threshold and at least six times to reach the 0.2% threshold. The observation that read coverage was not correlated with the proportion of positions with unusual mutations also supports the conclusion that most unusual mutations did not result from machine error.

APOBEC-mediated G-to-A hypermutation is not a result of PCR error and it presents in sequences even when PCR errors are excluded through the use of unique molecular identifiers (UMIs) [9]. This study indicates that, at the thresholds of 0.5%, 1%, and 2%, signature APO-BEC mutations outnumber non-APOBEC unusual mutations in approximately one-sixth of samples even though non-APOBEC unusual mutations are far more numerous than signature APOBEC mutations. There are 17 DRMs that could be caused by APOBEC-mediated G-to-A hypermutation: D30N, M46I, G48S, and G73S in PR, D67N, E138K, M184I, G190ES, and M230I in RT, and G118R, E138K, G140S, G163KR, D232N, and R263K in IN. These mutations should be considered possible artifacts if they occur at the same threshold at which multiple signature APOBEC mutations are also present.

To estimate the proportions of positions with unusual mutations generated during HIV-1 replication *in vivo*, we recently performed a meta-analysis of publicly available *pol* single genome sequences (SGSs)–which are not subject to PCR error–in plasma samples from persons with active HIV-1 replication [8]. We found that in samples with a median of 20 SGSs, the proportion of sequence positions with an unusual mutation was ≤1% in 90% of samples

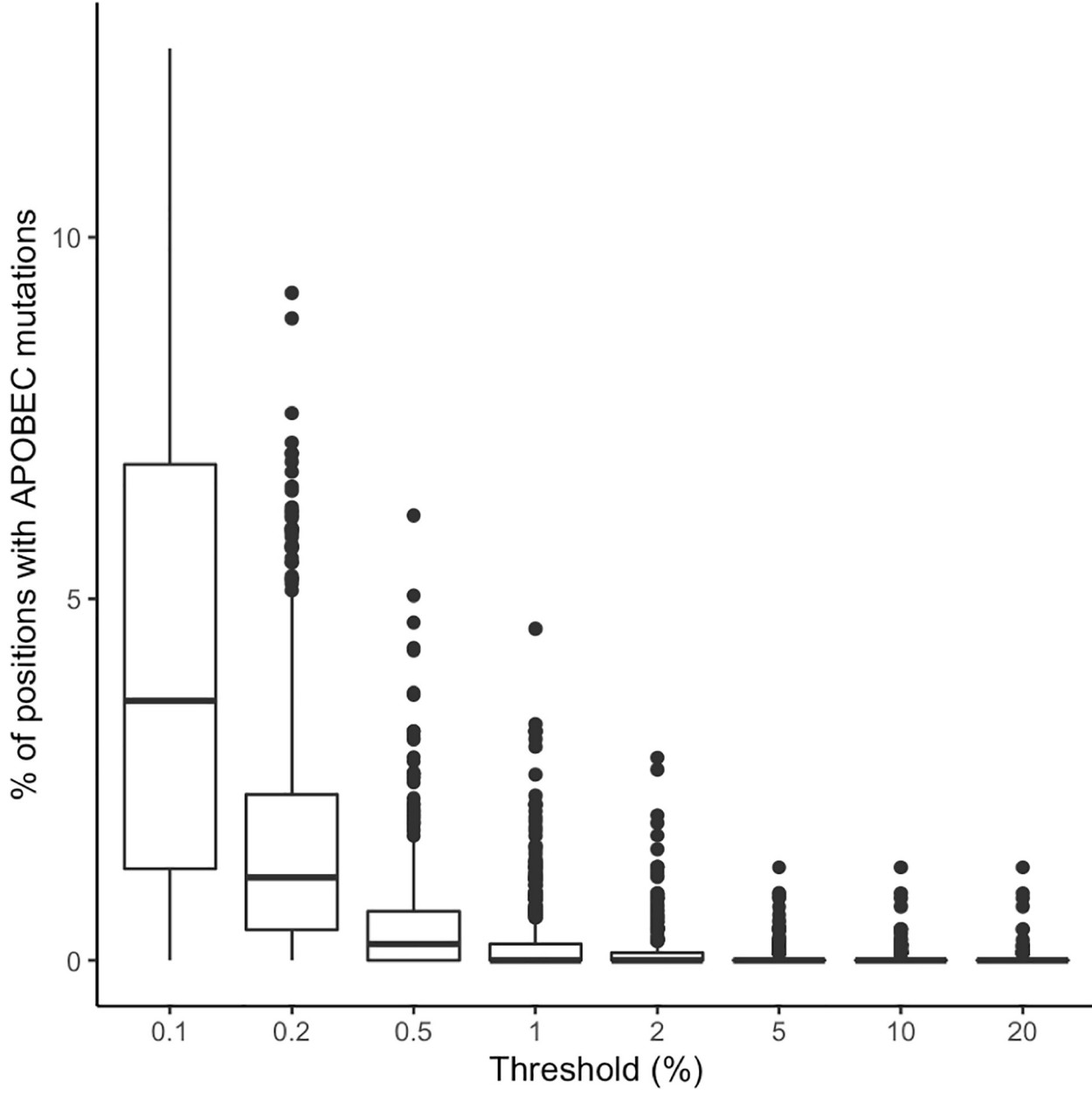

**Fig 5. Boxplots demonstrating the distribution in the proportion of positions with signature APOBEC mutations at eight NGS mutation detection thresholds for pooled samples (n = 855) from eight published studies.**

and ≤3% in 99% of samples. Similarly, the proportion of all mutations that were unusual was ≤15% in 90% of samples and ≤33% in 99% of samples. Although the proportion of positions with an unusual mutation might have been higher had the median number of SGSs been >20, the proportion of mutations that were unusual would not be affected by the median number of SGSs. Additionally, the estimates of the expected number of unusual mutations based on these SGS data are likely to be inflated as SGSs are subject to errors when RNA is reversed transcribed to cDNA.

The problem of PCR error, as well as the problems of PCR recombination and biased amplification, have led to an approach in which UMIs are added to each cDNA molecule

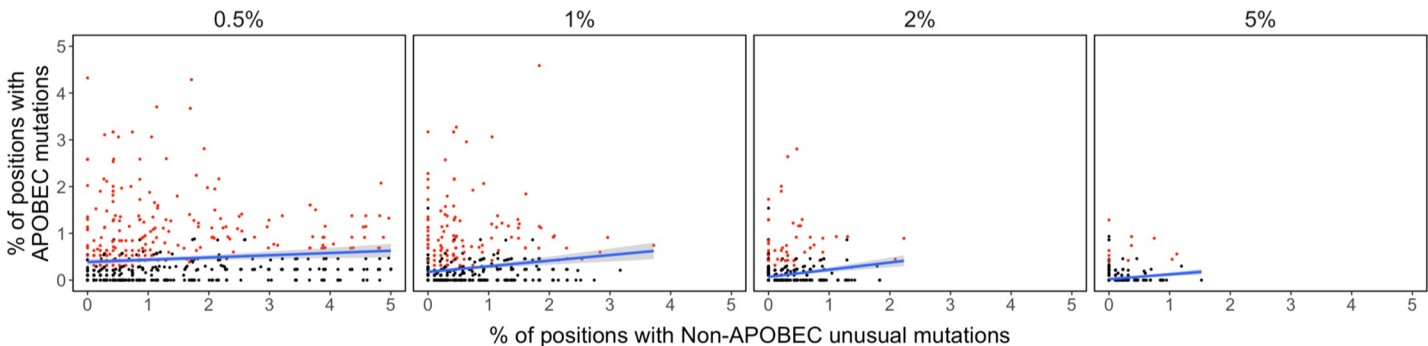

**Fig 6. Scatterplots demonstrating the relationship between the number of signature APOBEC mutations and the number of unusual mutations that were not signature APOBEC mutations at four NGS mutation detection thresholds.** Samples containing three or more signature APOBEC mutations are shown in red.

during reverse transcription [29]. Following PCR, consensus sequences are created from reads having the same random ID, making it possible to identify and exclude these errors, which present as mutations that differ from the consensus of sequences containing the same UMI. This approach also has the advantage of providing an estimate of the number of successfully amplified cDNA molecules and of identifying non-consensus sequences that result from PCR recombination. The use of UMIs is becoming the standard approach for investigating HIV-1 dynamics *in vivo*. However, the use of UMIs necessitates multiple RT reactions and PCR amplifications to generate amplicons spanning HIV-1 drug targets, thus increasing both expense and complexity for genotypic resistance testing for routine clinical or surveillance testing.

Several other approaches can also reduce the risk of PCR errors or the likelihood that such errors will lead to artifactual results. First, PCR errors can be reduced by using high fidelity PCR enzymes, although such enzymes often have reduced processivity [30]. Second, the number of PCR cycles may be reduced if a sufficient number of genomic templates are present prior to PCR. Third, co-occurrence of low abundance variants can be used to increase the confidence of each constituent mutation, although such co-occurrences can only be observed over the short genetic distances because most sequence reads are short [31]. Although laboratories often estimate the probability of PCR errors by sequencing plasmid controls, PCR error rates determined by this approach are often underestimates, particularly on samples from which only a small number of cDNA molecules can be generated, as these samples will undergo more PCR amplification than plasmid DNA controls.

Our analysis was not designed to recommend a single NGS mutation detection thresholds for routine clinical genotypic resistance testing. First, although the one percent threshold was generally associated with low numbers of unusual and signature APOBEC mutations, a subset of samples had high numbers of these mutations above the one percent threshold. Second, the analyzed data was generated by research laboratories that adopted measures to improve the detection of low frequency variants, avoided samples with very low virus loads, and made their data publicly available. Indeed, previous studies have suggested that a five percent threshold is likely to be more reproducible in clinical settings [7, 24, 32, 33].

In conclusion, we propose that that a *post hoc* analysis of HIV-1 pol NGS data that quantifies the numbers of unusual and signature APOBEC mutations at different NGS mutation detection thresholds can be useful to avoid selecting a threshold that is too low and that poses an unacceptable risk of identifying artifactual mutations. Such an analysis demonstrates how the body of published sequence data for a gene (i.e., the prevalence of all mutations at all positions), can be used to optimize the analysis of a new sequence of that gene. To this end, we

created a version of the HIVDB genotypic resistance interpretation program that accepts HIV-1 *pol* codon frequency tables and provides counts of unusual and signature APOBEC mutations and genotypic resistance interpretations at user-selected mutation detection thresholds (https://hivdb.stanford.edu/hivdb/by-reads/).

## Supporting information

**S1 Text. Comparison of the number of signature APOBEC mutations in a sequence with the results of the Los Alamos National Laboratories (LANL) HIV Sequence Database Hypermut2 program.**
(DOCX)

**S1 Fig. Scatter plots demonstrating the relationship between log 10 sequence reads and the proportion of positions with usual mutations.** Each plot contains the Pearson correlation coefficient (r) and its associated p value.
(TIFF)

**S2 Fig. Scatter plots demonstrating the relationship between log 10 sequence reads and the proportion of positions with unusual mutations.** Each plot contains the Pearson correlation coefficient (r) and its associated p value.
(TIFF)

**S3 Fig. Median and 95% confidence intervals for the proportion of positions with unusual mutations at eight NGS mutation detection thresholds for each of the eight studies: A [23], B [19], C [24], D [20], E [25], F [18], G [21], H [22].**
(TIFF)

**S4 Fig. Scatter plot demonstrating the relationship between virus load (plasma HIV-1 RNA log copies/ml) and the proportion of positions with usual mutations at four NGS mutation detection thresholds in two of the three studies for which virus load data were available.**
(TIFF)

## Author Contributions

**Conceptualization:** Sergei L. Kosakovsky Pond, Robert W. Shafer.

**Data curation:** Philip L. Tzou, Santiago Avila-Rios.

**Formal analysis:** Philip L. Tzou, Robert W. Shafer.

**Funding acquisition:** Robert W. Shafer.

**Investigation:** Philip L. Tzou, Robert W. Shafer.

**Methodology:** Philip L. Tzou, Sergei L. Kosakovsky Pond, Robert W. Shafer.

**Software:** Philip L. Tzou.

**Supervision:** Susan P. Holmes.

**Validation:** Santiago Avila-Rios, Rami Kantor.

**Writing – original draft:** Robert W. Shafer.

**Writing – review & editing:** Sergei L. Kosakovsky Pond, Santiago Avila-Rios, Susan P. Holmes, Rami Kantor.

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
