## [Decision Letter · Decision Letter 0]

7 Jan 2020

PONE-D-19-30406

Analysis of unusual and signature APOBEC-mutations in HIV-1 pol next-generation sequences

PLOS ONE

Dear Mr. Tzou,

Thank you for submitting your manuscript to PLOS ONE. After careful consideration, we feel that it has merit but does not fully meet PLOS ONE’s publication criteria as it currently stands. Therefore, we invite you to submit a revised version of the manuscript that addresses the points raised during the review process.

We would appreciate receiving your revised manuscript by Feb 21 2020 11:59PM. To enhance the reproducibility of your results, we recommend that if applicable you deposit your laboratory protocols in protocols.io, where a protocol can be assigned its own identifier (DOI) such that it can be cited independently in the future. For instructions see: http://journals.plos.org/plosone/s/submission-guidelines#loc-laboratory-protocols

We look forward to receiving your revised manuscript.

Kind regards,

Orna Mor

Academic Editor

PLOS ONE

Journal Requirements:

Additional Editor Comments (if provided):

I find your submited paper "Analysis of unusual and signature APOBEC-mutations in HIV-1 pol next-generation sequences" well written, interesting and relevant to the field of HIV and other NGS- Illumina derived data. Both reviewers agreed that this paper is worth publishing with some minor revisions. I look forward to reading this paper in pubmed.

Reviewers' comments:

Reviewer's Responses to Questions

**Comments to the Author**

1. Is the manuscript technically sound, and do the data support the conclusions?

Reviewer #1: Yes

Reviewer #2: Yes

2. Has the statistical analysis been performed appropriately and rigorously? 

Reviewer #1: Yes

Reviewer #2: N/A

3. Have the authors made all data underlying the findings in their manuscript fully available?

Reviewer #1: Yes

Reviewer #2: Yes

4. Is the manuscript presented in an intelligible fashion and written in standard English?

Reviewer #1: Yes

Reviewer #2: Yes

5. Review Comments to the Author

Reviewer #1: In this paper, a meta-analysis of already-published HIV gene sequences was performed to assess the number of usual, unusual and APOBEC-generated mutations at different NGS mutation detection thresholds. The authors list all the different types of mutations they found and suggest to quantify the numbers of unusual and APOBEC-generated mutations at different detection thresholds to avoid identification of artificial mutations, and provide a tool to assess these mutations in HIVDB.

Minor comments

- “NGS thresholds” (e.g. in the abstract, or at the end of the introduction at page 5) is a little vague, might consider changing to “NGS mutation detection threshold” throughout the manuscript, where applicable.

- line 139, change “were” to “was” and “their” to “its”; line 140 change “their” to “its”

- line 149, “overall we identified 296 signature APOBEC mutations” - is this for all 3 genes? how many in each? (e.g. pol gene)

- there aren’t any explanations as to how the NGS mutation detection thresholds was determined/calculated. Should be added to “methods”.

- from the first paragraph of the discussion: “The marked increase in the proportion of positions with unusual mutations and in the proportion of mutations that were unusual at detection thresholds below 1% suggests that many of the mutations at low thresholds resulted from processes other than virus replication such as PCR error and APOBEC-mediated hypermutation”. why? this statement should at least be explained and preferably backed up by facts from the literature re HIV mutation rate, etc.

- line 287, might consider using the term “depth” rather than coverage to be more clear as to how many repeats were observed for each position.

Reviewer #2: The manuscript describes a well-conducted study that is of clinical importance for HIV-1 resistance studies based on NGS technology.

My only comment is if authors can make a comment/recommendation regarding the NGS detection threshold to be used in routine clinical practice

6. PLOS authors have the option to publish the peer review history of their article (what does this mean?). If published, this will include your full peer review and any attached files.

Reviewer #1: No

Reviewer #2: Yes: Dimitrios Paraskevis

---

## [Author Response · Author response to Decision Letter 0]

21 Jan 2020

Reviewer #1: In this paper, a meta-analysis of already-published HIV gene sequences was performed to assess the number of usual, unusual and APOBEC-generated mutations at different NGS mutation detection thresholds. The authors list all the different types of mutations they found and suggest to quantify the numbers of unusual and APOBEC-generated mutations at different detection thresholds to avoid identification of artificial mutations, and provide a tool to assess these mutations in HIVDB.

Minor comments:

 “NGS thresholds” (e.g. in the abstract, or at the end of the introduction at page 5) is a little vague, might consider changing to “NGS mutation detection threshold” throughout the manuscript, where applicable. We changed “NGS threshold” to “NGS mutation detection threshold” or “threshold” to “mutation detection threshold” at 12 locations in the manuscript – wherever either of these terms first appeared for example in the header of a new section, the start of a paragraph, or in a figure legend. Specifically, these changes were made in the Abstract Methods and Conclusion section; the Introduction, the legends for Figures 2, 3, 4, and 5 and for Supplementary Figures 3 and 4; two section headers in the Results; and the concluding paragraph in the Discussion. There were many locations in which the term “threshold” was used that we left unchanged because the context was clear as a result of the changes outlined above. 

line 139, change “were” to “was” and “their” to “its”; line 140 change “their” to “its”. These changes were made. 

line 149, “overall we identified 296 signature APOBEC mutations” - is this for all 3 genes? how many in each? (e.g. pol gene). We changed “Overall, we identified 296 signature APOBEC mutations.” to “Overall, we identified 296 signature APOBEC mutations including 45 in PR, 154 in RT, and 97 in IN.” 

there aren’t any explanations as to how the NGS mutation detection thresholds was determined/calculated. Should be added to “methods”. We added a “Statistical analysis” section to the Methods: 

“Statistical analysis

 We calculated the proportion of amino acid positions with usual mutations, unusual mutations, and signature APOBEC mutations at eight NGS mutation detection thresholds. Usual mutations were defined as differences from the subtype B consensus sequence that were not unusual. We also calculated the proportion of all mutations that were unusual (number of unusual mutations / total number of mutations) at these same thresholds. The eight mutation thresholds began at 20%, which is often considered the limit of detection of mixed bases for Sanger sequencing, with each subsequent value approximately two-fold lower than the previous threshold: 10%, 5%, 2%, 1%, 0.5%, 0.2%, and 0.1%. Such “round” thresholds are commonly used in manuscripts performing NGS data interpretation, and are meant to serve as representative values spanning the realistic range used by researchers. 

Pearson correlation coefficient (r) was used to quantify the association between a sample’s (i) virus load and proportion of positions with usual or unusual mutations; and (ii) median number of sequence reads per position and proportion of positions with usual or unusual mutations.”

from the first paragraph of the discussion: “The marked increase in the proportion of positions with unusual mutations and in the proportion of mutations that were unusual at detection thresholds below 1% suggests that many of the mutations at low thresholds resulted from processes other than virus replication such as PCR error and APOBEC-mediated hypermutation”. why? this statement should at least be explained and preferably backed up by facts from the literature re HIV mutation rate, etc. We have added the following text following the first sentence (quoted above) in the second paragraph of the discussion: “Although HIV-1 RT produces approximately one random nucleic acid mutation per genome each replication cycle [26, 27], the fitness costs of most random nonsynonymous mutations are high. Therefore, viruses containing many, if not most, random mutations are rarely compatible with levels of replication that would yield viruses detectable at levels significantly higher than the background error rate [28]. In contrast, PCR errors and APOBEC-mediated G-to-A hypermutation can result in mutations detectable at levels that would not be consistent with virus replication.”

line 287, might consider using the term “depth” rather than coverage to be more clear as to how many repeats were observed for each position. In the Methods section we indicate that “coverage” refers to “the number of reads per position”. We would prefer not to change “coverage” to “depth” because it seems to us that “depth” connotes that the more reads that are present at a position, the more insight is gained into the HIV-1 quasispecies. Furthermore, the term “coverage”, is standard in genomic literature, and is used to describe the number of sequencing reads successfully mapped to a genomic position; it is good practice to use standard terminology whenever possible. Perhaps occasional use of “depth” relates to the frequent use of the term “ultra-deep sequencing” in the past. In contrast, we believe that “coverage” is a more neutral term. This is particularly relevant to the manuscript because one of its main points is that the number of reads at a position does not guarantee that the data at low mutation thresholds are reliable. 

Reviewer #2: The manuscript describes a well-conducted study that is of clinical importance for HIV-1 resistance studies based on NGS technology. My only comment is if authors can make a comment/recommendation regarding the NGS detection threshold to be used in routine clinical practice. We have added the following paragraph before the concluding paragraph of the Discussion: “Our analysis was not designed to recommend a single NGS mutation detection thresholds for routine clinical genotypic resistance testing. First, although the one percent threshold was generally associated with low numbers of unusual and signature APOBEC mutations, a subset of samples had high numbers of these mutations above the one percent threshold. Second, the analyzed data was generated by research laboratories that adopted measures to improve the detection of low frequency variants, avoided samples with very low virus loads, and made their data publicly available. Indeed, previous studies have suggested that a five percent threshold is likely to be more reproducible in clinical settings [7, 24, 32, 33].”

---

## [Editor Report · Decision Letter 1]

31 Jan 2020

Analysis of unusual and signature APOBEC-mutations in HIV-1 pol next-generation sequences

PONE-D-19-30406R1

Dear Dr. Tzou,

We are pleased to inform you that your manuscript has been judged scientifically suitable for publication and will be formally accepted for publication once it complies with all outstanding technical requirements.

With kind regards,

Orna Mor

Academic Editor

PLOS ONE
---

## [Editor Report · Acceptance letter]

11 Feb 2020

PONE-D-19-30406R1 

Analysis of unusual and signature APOBEC-mutations in HIV-1 pol next-generation sequences 

Dear Dr. Tzou:

I am pleased to inform you that your manuscript has been deemed suitable for publication in PLOS ONE. Congratulations! Your manuscript is now with our production department. 

With kind regards,

on behalf of

Dr. Orna Mor 

Academic Editor

PLOS ONE